# Does the Element Availability Change in Soils Exposed to Bioplastics and Plastics for Six Months?

**DOI:** 10.3390/ijerph19159610

**Published:** 2022-08-04

**Authors:** Giorgia Santini, Giulia Maisto, Valeria Memoli, Gabriella Di Natale, Marco Trifuoggi, Lucia Santorufo

**Affiliations:** 1Department of Biology, University of Naples Federico II, Via Cinthia, 80126 Naples, Italy; 2BAT Center—Center for Studies on Bioinspired Agro-Environmental Technology, 80126 Naples, Italy; 3CeSMA—Centre of Meteorologic and Avanced Thecnology Services, University of Naples Federico II, Nicolangelo Protopisani Course, San Giovanni a Teduccio, 80146 Naples, Italy

**Keywords:** microplastic, metal contamination, polyethylene, biodegradable plastic, soil contamination

## Abstract

Plastic sheets are widely used in farming soil to improve the productivity of cultures. Due to their absorption capacity, plastic sheets can alter element and metal content in soils, and in turn affect soil properties. The use of biodegradable films is an attractive eco-sustainable alternative approach to overcome the environmental pollution problems due to the use of plastic films but their impacts on soil are scarcely studied. The aim of the research was to evaluate the impact of conventional plastic and bioplastic sheets on total and available concentrations of elements (Al, Ca, Cu, Fe, K, Mg, Mn, Na, Ni, Pb, and Zn) in soils. The research was performed in mesocosm trials, filled with soil covered by conventional plastic and bioplastic sheets. After six months of exposure, soils were characterized for pH, water content, concentrations of organic and total carbon and total nitrogen, and total and available Al, Ca, Cu, Fe, K, Mg, Mn, Na, Ni, Pb, and Zn element concentrations. The results highlighted that soils covered by bioplastic sheets showed higher total and available concentrations of elements and higher contamination factors, suggesting that bioplastic sheets represented a source of metals or a less-effective sink to these background metals in soils, compared to conventional plastic ones.

## 1. Introduction

The huge production of plastic materials has caused a widespread dispersion of plastic waste into the environment, forming debris of microplastics (MP) with size ranging from 0.1 to 5 mm [1,2]. As MPs have long persistence and slow degradation, they are ubiquitous in the environment and are recognized as emergent pollutants [3], that can cause serious hazards to organisms. The presence of MPs and their effects have been widely investigated in the aquatic environment, but the research in the terrestrial environment is incomparably lacking.

Agricultural soils can be polluted by MPs owing to intensive human activities such as application of sewage sludge and compost and fragmentation of plastic mulches [4]. As agroecosystems provide food, MPs in soils could cause unknown effects on farm ecosystems and food security, posing serious risks for human health [5]. Considering the risks microplastics pose to the ecosystem through the food chain, it is essential to understand the behavior of microplastics in the agricultural soil systems. The effects and the fate of MPs in soils are still controversial. It is well known that MPs can change soil porosity, water retention, and bulk density [6]. Moreover, they can be also responsible for the release of harmful additives [7] that can worsen the overall soil quality [8,9].

Polyethylene-MPs are the main kind of MPs in soil environments [10] and are also the main material of agricultural film, which is widely utilized in farming soil [11,12]. MPs directly or indirectly affect soil ecosystem functions and microbial communities [13,14,15], changing soil structure [6,16], soil pH [17,18,19], and increased soil aggregation [20]. Moreover, MPs can alter nutrient and metal content in soils [21,22], by absorbing some contaminants, such as Zn [23,24]. In addition, MPs, by modifying the soil abiotic properties can indirectly influence the chemical forms and bioavailability of heavy metals [11,25,26]. Therefore, MPs are an important factor governing the transformation of heavy metal speciation in soil. Although some studies have shown that MPs can adsorb heavy metals [23,27,28], potential changes in the chemical speciation of heavy metals triggered by MP contamination have scarcely been studied. Moreover, soil is a matrix with diverse microenvironments rather than a homogenous matrix [29], and this can lead to components and environments in different soil fractions responding differently to changes in the external environment. This response mechanism warrants investigation given its expected importance in guiding soil management.

The use of biodegradable films is an attractive eco-sustainable alternative approach to overcome the environmental pollution problems due to the use of plastic films [30]. Biodegradable films have already been tested as soil mulching films on several crops, such as zucchini squash [31], tomato [32], strawberry [33], lettuce [34], pepper, eggplant, musk melon, and sweet corn [35]. However, the impact of biodegradable plastic application to soil is not deeply investigated, and their effects on soil characteristics and element bioavailability are scarcely studied. Current studies on the differences between the adsorption capacities of bioplastics and conventional plastics for chemical pollutants have not yet reached an unambiguous conclusion.

Therefore, the aim of the present research was to evaluate the impact of conventional plastic and bioplastic sheets on total and available concentrations of elements (Al, Ca, Cu, Fe, K, Mg, Mn, Na, Ni, Pb, and Zn) in soils. To achieve the aim, the research was performed in mesocosm trials in which conventional plastic and bioplastic sheets were placed on soils and the effects were evaluated after six months since exposure.

## 2. Materials and Methods

### 2.1. Mesocosm Setting Up

The experiment was performed in mesocosms. Ten pots, of one meter in diameter, were filled to 40 cm of height with limestone debris of different granulometry (1–4 cm diameters) picked up in a quarry located near Caserta. Contextually, in November 2020, forest surface soil was collected inside the Natural Reserve of Astroni. Approximately 50 kg of soil were placed on the limestone debris of each of the 10 pots for 30 cm of height.

In December 2020, a sheet (40 × 40 cm) of conventional plastic (made by polyethylene) constituted by little 16 squares (10 × 10 cm) was placed on the surface of the soil of five pots; whereas a sheet of bioplastic (made from polysaccharide complexes) of the same size was placed on the surface of the soil of five other pots. The choice of the materials was made according to the widespread plastic and emergent bioplastic mulches used in southern Italy in agriculture. The mesocosms were left outdoors on the terrace of the Department of Biology of the University of Naples Federico II.

### 2.2. Sampling and Analyses

In January 2021, before the placement of sheets of plastics (T0), surface soils (0–10 cm) were collected from each of the 10 pots, sieved (mesh: 2 mm) and characterized for pH, water content, concentrations of organic and total carbon, and total nitrogen (Table 1). Moreover, total and available element concentrations (Al, Ca, Cu, Fe, K, Mg, Mn, Na, Ni, Pb, and Zn) were evaluated.

Six months (T2: July 2021) after the mesocosm was set up, cores of soils were collected under a little square of the plastic sheet (10 × 10 cm) from each of the 10 pots by a sampler (10 cm Ø) from the upper 10 cm layer. The soil samples were analyzed for the same properties detected at T0.

The total concentrations of Al, Ca, Cu, Fe, K, Mg, Mn, Na, Ni, Pb, and Zn were measured in dried soil samples (80 °C), pulverized by an agate mortar (Fritsch Analysette Spartan 3 Pulverisette 0), and digested by hydrofluoric acid (50%) and nitric acid (65%) in a ratio of 1:2 (v:v) in a microwave oven (Milestone-Digestion/Drying Module mls 1200). According to the method of Lindsay and Norwell [36], the available fractions of Al, Ca, Cu, Fe, K, Mg, Mn, Na, Ni, Pb, and Zn were extracted. In brief, 25 mL of pentacetic diethylentriaminic acid (DTPA), CaCl2, and triethanolamine (TEA) solution at pH 7.3 ± 0.05 was added to 12.5 g of oven-dried soil samples (75 °C) to measure the Al, Cu, Fe, Mn, Ni, Pb, and Zn fractions. Whereas the availability of Na, Mg, K, and Ca was evaluated by BaCl_2_ and TEA pH 8.1 [36]. The soil suspensions were stirred for 2 h and filtered through a Whatman 42 filter. Element concentrations in the digests and extracts were measured by inductively coupled plasma mass spectrometry (ICP-MS Aurora M90, Bruker, Germany).

### 2.3. Assessment of Soil Metal Contamination

The Contamination Factor (CF) index was calculated in order to evaluate the contamination extent of the investigated soil for each metal (Al, Cu, Fe, Mn, Ni, Pb, and Zn); whereas the Pollution Load Index (PLI) was calculated in order to evaluate the metal whole and integrated contamination extent of the investigated soils.

The CF was calculated as reported below [37,38]:(1)$$\mathrm{CF}=\frac{C}{\mathrm{Bn}}$$
where C represents the concentration of the metal in soil samples and Bn represents the background value of the same metal in soils of the Campania Region [39]. Luo et al. [37] have distinguished the CF into four classes: CF < 1, low contamination factor; 1 ≤ CF < 3, moderate contamination factors; 3 ≤ CF < 6, considerable contamination factors; and C ≥ 6, very high contamination factor.

The PLI was calculated as reported below [38,40]:(2)$$\mathrm{PLI}=\sqrt[{n}]{\prod_{i=1}^{n}\mathrm{CF}}$$
where n is the metal and CF is the contamination factor. Banu et al. [41] have distinguished the PLI into two classes: PLI < 1, no pollution and PLI > 1, pollution.

### 2.4. Statistical Analyses

The normality of the dataset distribution passed (Wilk–Shapiro test for α = 0.05; n = 14); therefore, parametric tests were performed.

Student *t*-tests were performed to evaluate the differences in metal concentrations and in the calculated CF and PLI in soils exposed to conventional plastic and bioplastic and was considered significant at least for α < 0.05.

The statistical analyses were performed by using the R 4.0.3 programming environment. The graphs were created using the SigmaPlot12 software (Jandel Scientific, San Rafael, CA, USA).

## 3. Results

### 3.1. Soil Total Metal Concentrations

The total concentrations of the investigated metals at the beginning of experiment (T0) and those in soils covered by both types of plastics are reported in Figure 1. The comparison between soils covered by bioplastic and conventional plastic highlighted that the concentrations of K, Mn, Na, Ni, Pb, and Zn did not significantly vary (Figure 1); by contrast, those of Al, Ca, Fe, Mg, and Cu were significantly higher in soils covered by bioplastic (70.8, 62.4, 31.5, 10.1 mg g^−1^ d.w., and 121 μg g^−1^ d.w., respectively, for Al, Ca, Fe, Mg, and Cu) than in those covered by conventional plastic (57.5, 37.2, 26.1, 7.5 mg g^−1^ d.w., and 101 μg g^−1^ d.w., respectively, for Al, Ca, Fe, Mg, and Cu) (Figure 1).

The Contamination Factors (CFs), reported in Table 2, were statistically higher in soils covered by bioplastic for Al, Fe, and Mg than in those covered by conventional plastic.

The PLI values were 1.11 and 1.26, respectively, for soils covered by conventional plastic and bioplastic, with no significant differences between the treatments (Table 2).

### 3.2. Soil Available Fractions

The available fractions of the investigated metals at the beginning of experiment are reported in Appendix A, and those in soils covered by both types of plastic are reported in Figure 2. The comparison between soils covered by bioplastic and conventional plastic highlighted that the availabilities of Fe, Mg, Mn, Na, Ni, and Zn were significantly higher in soils covered by bioplastic (0.45, 0.79 mg g^−1^ d.w., and 34.1, 7.26, 0.26, 28.5 μg g^−1^ d.w., respectively, for Mg, Na, Fe, Mn, Ni, and Zn) than in those covered by conventional plastic (0.392, 0.69 mg g^−1^ d.w., and 29.8, 6.31, 0.182, 27.4 μg g^−1^ d.w., respectively, for Mg, Na, Fe, Mn, Ni, and Zn) (Figure 2). By contrast, the availability of Pb was significantly higher in soils covered by conventional plastic (8.09 and 7.24 μg g^−1^ d.w., respectively, in soils covered by conventional plastic and bioplastic) (Figure 2).

### 3.3. Percentages of Metal Availability with Respect to Total Concentration

The percentages of metal availability with respect to total concentration of Al, Ca, Cu, Mg, and Pb were significantly higher in soils covered by conventional plastic (0.003, 13.1, 42.2, 5.91, 9.23%, respectively, for Al, Ca, Cu, Mg, and Pb), than in those covered by bioplastic (0.002, 8.83, 36.1, 4.61, 8.00%, respectively, for Al, Ca, Cu, Mg, and Pb); by contrast, that of Zn was higher in soils covered by bioplastic (20.6 and 18.5%, respectively, for soils covered by bioplastic and conventional plastic) (Table 3).

## 4. Discussion

The findings highlighted that the presence of bioplastic sheets on soils caused significant increases in total and available concentrations of most of the investigated metals. This is corroborated by the observed capability of plastics to hold metals [27,42], which may be added as pigments or heat stabilizers during different phases of their production [43]. Moreover, the capability of plastics to hold metals depends also on their characteristics, such as pore filling, hydrophobic interactions, hydrogen bonds, electrostatic interactions, van der Waals forces, and specific surface area [23,44].

The significantly higher total concentrations of Al, Ca, Cu, Fe, and Mg in soils covered by bioplastic than conventional plastic sheets likely can be due to the nature of the bioplastic itself. In fact, the used bioplastic sheets, constituted by polysaccharide complexes such as amylose and amylopectin, likely showed many O-ligands to link metal ions [45]. Bioplastics, because of their crystallization characteristics and carrier adsorption, have stronger adsorption capacity of metals than conventional plastic [10,46].

The higher degradability of bioplastic sheets would seem to be responsible for the greater availability of Fe, Mg, Mn, Na, Ni, and Zn observed in soils covered by bioplastic than by conventional plastic sheets. In fact, the used bioplastic sheets, rich in organic carbon compounds, represent an important resource for the soil-dwelling microorganisms [47,48]. The formation of fragments of bioplastic sheets, due to biological activity, increased the surface of contact with soils, enhancing the further physical and/or chemical fragmentation of the sheet itself [10]. Although the availability of most of the investigated elements were higher in soils covered by bioplastic sheets, Pb availability was meaningfully higher in soils covered by conventional plastic sheets. Likely, this could be due to the release from conventional plastic sheets of petroleum-based compounds added during their production [43].

Despite bioplastic being widely regarded as an environmentally friendly substitute for conventional plastic, its effects on metal accumulation in soils remain largely unknown. Unfortunately, the PLI (greater than 1) highlighted that the bioplastic sheets represented sources of metal contamination for the soils to the same extent as the conventional plastic sheets [23]. Moreover, bioplastic was responsible for the greater accumulation of Al, Fe, and Mg, as the contamination factors for these elements were significantly higher than in soils covered by the conventional plastic sheets.

The percentage of the available fraction of the element in relation to its total concentration showed significant differences in soils exposed to the investigated sheets. These percentages calculated for Al, Ca, Cu, Mg, and Pb were significantly higher in soils exposed to conventional plastic sheets, suggesting that these sheets decreased the adsorption capability of elements to soil and increased their desorption [49]. Instead, the stronger adsorption capacity of bioplastic sheets [10] increased their degradation rate, causing higher total and available concentrations of the elements. The lack of significant differences for the other investigated elements could be due to the fact that, likely, they are linked in chemical complexes that need a longer time to be exchanged.

## 5. Conclusions

The overall comparison between soils covered by conventional plastic and bioplastic sheets highlighted that soils covered by the former showed higher ratios between available and total concentrations of elements, whereas those covered by the latter showed higher total and available concentrations of elements and higher contamination factors. Both the kinds of plastic sheets caused soil metal accumulation as the pollution load indices were greater than 1.

Thereby, the findings suggest that bioplastic sheets represented a source of metal contamination or a less-effective sink to these background metals, comparable to the conventional plastic ones.

Further investigations are needed to evaluate the effects of metal accumulation in soils due to longer periods of plastic sheet exposure.

## Figures and Tables

**Figure 1 ijerph-19-09610-f001:**
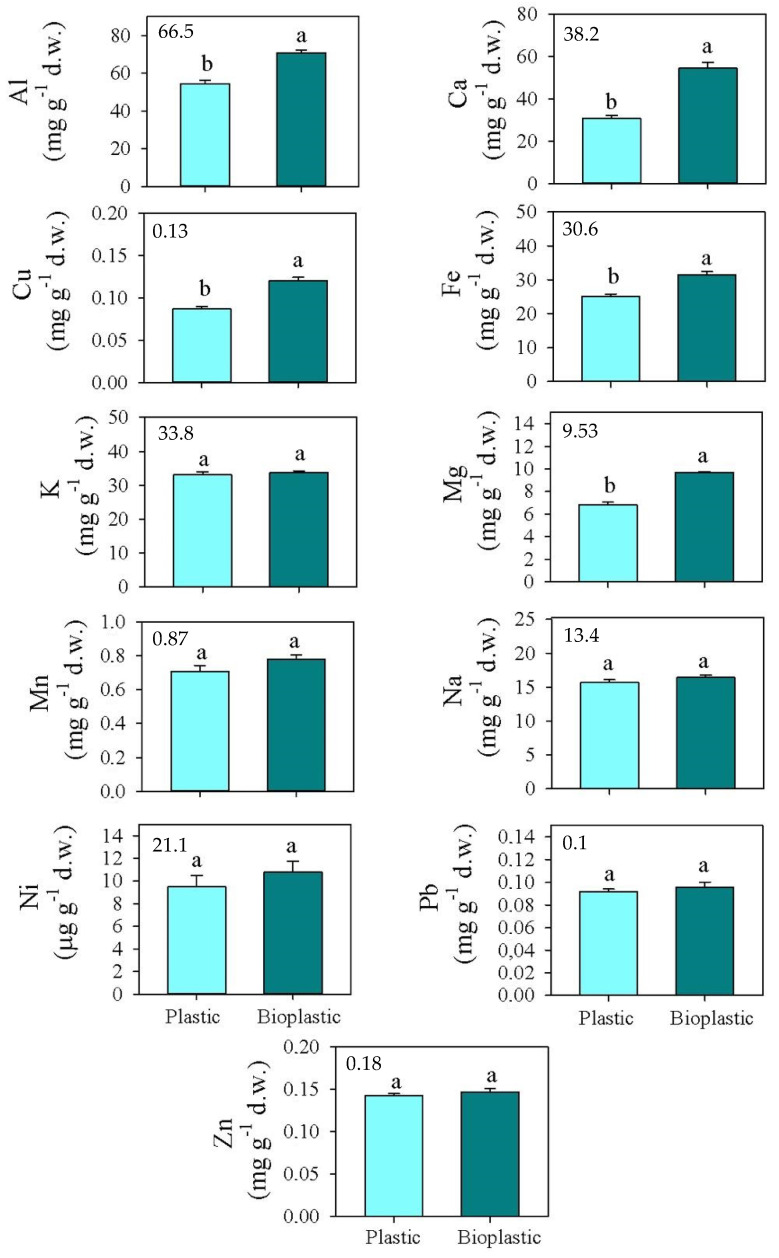
Mean values (±s.e.) of total Al, Ca, Cu, Fe, K, Mg, Mn, Na, Pb, and Zn (expressed as mg g^−1^ d.w.), and Ni (expressed as μg g^−1^ d.w.) concentrations measured in soils at T0 (**top left**) and after six months of exposure to plastic (light blue) and bioplastic (dark blue). Different small letters indicate statistically significant differences between soils covered by plastic and bioplastic, respectively (one-way ANOVA; *p* < 0.05).

**Figure 2 ijerph-19-09610-f002:**
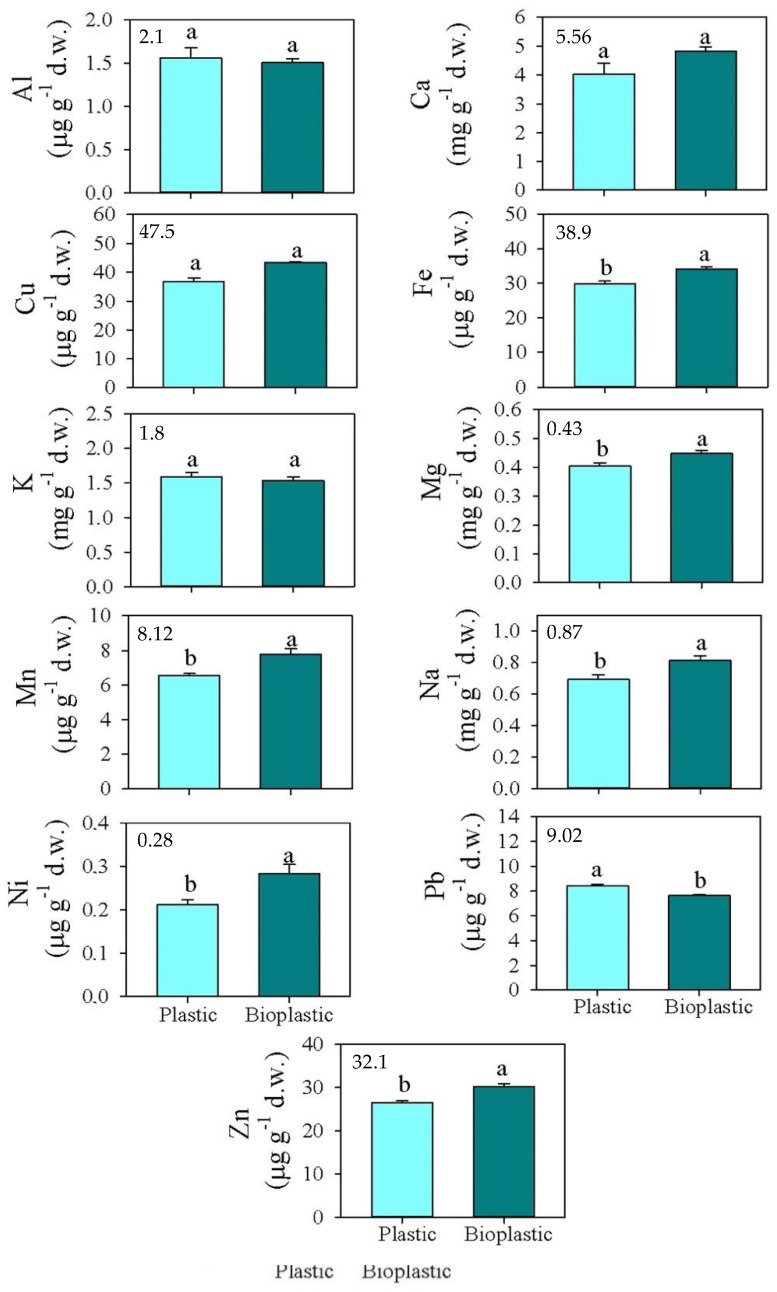
Mean values (±s.e.) of available Al, Cu, Fe, Mn, Pb, and Zn (expressed as μg g^−1^ d.w.), and Ca, K, Mg, and Na (expressed as mg g^−1^ d.w.) concentrations measured in soils at T0 (**top left**) and after six months of exposure to plastic (light blue) and bioplastic (dark blue). Different small letters indicate statistically significant differences between soils covered by plastic and bioplastic, respectively (one-way ANOVA; *p* < 0.05).

**Table 1 ijerph-19-09610-t001:** Mean values ± s.e. (n = 14) of soil properties (pH; water content: WC (% d.w.); C total concentration (% d.w.); N total concentration (% d.w.); organic carbon concentration: C_org_ (% d.w.); C/N ratio (% d.w.)) at the beginning of the experiment (T0).

Abiotic Properties	Mean Values ± s.e.
pH	7.4 ± 0.07
WC	39.4 ± 0.63
C	4.2 ± 0.10
N	0.4 ± 0.01
C_org_	3.2 ± 0.05
C/N	10.8 ± 0.64

**Table 2 ijerph-19-09610-t002:** Mean values of the Contamination Factor (CF) and, in bold, of the Pollution Load Index (PLI) for the investigated soil covered by plastic and bioplastic. Asterisks indicate significant differences between plastic and bioplastic (*t*-test; *p* < 0.05).

	CF	
Metals	Plastic	Bioplastic
Al	1.47	1.81 *
Ca	2.12	2.51
Cu	0.62	0.74
Fe	1.10	1.33 **
K	1.55	1.58
Mg	1.29	1.74 *
Mn	0.96	1.02
Na	2.57	2.69
Ni	0.78	1.11
Pb	1.05	0.98
Zn	1.68	1.28
**PLI**	**1.11**	**1.26**

* *p* < 0.05; ** *p* < 0.01.

**Table 3 ijerph-19-09610-t003:** Mean values of ratios (expressed as percentage) of elements available and total concentration measured after six months of exposure. Asterisks indicate statistically significant differences between soils covered by plastic and bioplastic, respectively (*t*-test; *p* < 0.05).

Metals	Plastic	Bioplastic
Al	0.003	0.002 **
Ca	13.1	8.83 **
Cu	42.2	36.1 *
Fe	0.12	0.11
K	4.80	4.56
Mg	5.91	4.61 **
Mn	0.93	1.00
Na	4.41	4.96
Ni	2.23	2.64
Pb	9.23	8.00 *
Zn	18.5	20.6 **

* *p* < 0.05; ** *p* < 0.01.

## Data Availability

The data that has been used is confidential.

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
