# Peer review of "Does the Element Availability Change in Soils Exposed to Bioplastics and Plastics for Six Months?"

_ijerph, 2022, doi:10.3390/ijerph19159610_

Round 1

Reviewer 1 Report

It would be great to explain how the parameters in the ‘Mesocosm-setting-up’ are selected. For -example, number of pots, filled percentage, and soil amount.  Any reference that triggered this experiment setting?

How the author decided to use polyethylene conventional plastic and polysaccharide bioplastic for the comparison? Any underlying reasons for this choice? Do we think the result may change if we choose different conventional and bioplastic materials?

It would be interesting to see how the time affects the result. The paper demonstrated the result of six months. What if the experiment last longer (e.g. 8, or 12 months)? Will longer experiment time produces more distinctive result between conventional and bioplastics? 

In results section (e.g. table 3), it would be great to add some analysis or discussion on what might cause the different significance magnitude (e.g. 0.05 vs 0.01 vs not significant at all) among the selected elements.

+It’s great that the author showed the detailed significance test result for each element and provided explanation in the discussion section on what caused the element difference between conventional and bioplastics.

Author Response

The authors wish to thank the reviewer for the suggestions and comments,

Reviewer 2 Report

Santini et al.,

This manuscript presents experimental results demonstrating the difference between total and available metals in soil that were exposed to either bioplastics or plastics over six months.  While the scope of this study is limited, the authors appeared to have used a reliable, repeatable methodology to investigate their research question.  I only have a few key questions regarding the authors’ results and their interpretation.  I hope the authors find this feedback valuable.

General Comments

1.    I think more commentary regarding how the initial measured metals concentration varied with plastic and bioplastic sheets would be helpful.  To that end, perhaps the authors can include the background elements concentrations from the supplementary materials on Figures 1 and 2 to help the reader quickly visualize key changes?  This data appears to show that both plastic and bioplastic sheets may have decreased some total and available metals (e.g., Cu, Mn, Ni, Pb, Zn).  As the authors point out, this reduction may be due to absorption/adsorption.  However, I think the authors currently use background data from Cicchella et al., 2005 (10.1144/1467-7873/03-042) to discuss the contamination factors in Table 2, i.e., the changes in metals concentrations after the plastic/bioplastic sheets.  Why do the authors use this background data from Cicchella et al., 2005 as opposed to their own initial measurements?  

2.    It would be best to keep the units consistent throughout the paper.  

Specific Comments

1.    Lines 23-26: These two sentences are confusing, perhaps the authors can reword to draw the key conclusions?  The key conclusions are currently broken up in two sentences and it is unclear what the authors are comparing in the second sentence: “By contrast, soils covered by conventional plastic sheets showed an increase of available concentrations of elements”.

2.    Lines 26-27: Is it possible that bioplastics are not a source for some metals (e.g., Cu, Mn, Ni, Pb, Zn), but instead a less-effective sink to these background metals, compared to traditional plastic sheets?

3.    What exactly do ‘a’ and ‘b’ represent in Figures 1 and 2?

Author Response

(The authors gave the same response as above.)

Reviewer 3 Report

In my opinion, the introduction does not provide enough information regarding the problem of soil plastic pollution, it should be supplemented with information on how microplastics affect crop quality, transport of microplastics in soil. Whether and how MPs can change the physical, chemical and microbiological properties of soil. Such research has been conducted and described in these papers, among others:

FayuanWang et al. Effects of microplastics on soil properties: Current knowledge and future perspectives Journal of Hazardous Materials 2022. Lili Tian, Cheng Jinjin, Rong Ji, Yini Ma, Xiangyang Yu, Microplastics in agricultural soils: sources, effects, and their fate, Current Opinion in Environmental Science & Health, Volume 25, 2022,

Suggests expanding the introduction AND providing more information about the problem for potential readers.

Line 88, the authors give the number of pots as 14? Earlier in the text it was 10.

Why did the authors specifically choose these pollution indicators for their study? CF and PLI

Line 136 and 163, TableS1 ? what does S1 mean ?

Table 1 and Table 2 contain a lot of information and should be described more extensively by the authors.

The available fractions of the investigated metals at the beginning of experiment are 162 reported in Table S1, - there seems to be no such table in the text.

The article is indeed interesting, but the way of presenting the results is partly incomprehensible and not very clear

Author Response

(The authors gave the same response as above.)

Round 2

Reviewer 3 Report

The authors of the publication have complied with my suggested comments. The paper has been revised and is suitable for publication in its present form.